# Functional Analysis of Amino Acid Transporter Genes *ACYPI000536* and *ACYPI004320* in *Acyrthosiphon pisum*

**DOI:** 10.3390/insects15010020

**Published:** 2023-12-31

**Authors:** Lu Yao, Senshan Wang, Rui Ma, Jiangwen Wei, Liwen Song, Lei Liu

**Affiliations:** Gansu Provincial Crop Pest Biological Control Engineering Laboratory, College of Plant Protection, Gansu Agricultural University, Lanzhou 730070, China; yl1210856041@163.com (L.Y.); marui19990103@163.com (R.M.); wjw17883668289@163.com (J.W.); songlw@gsau.edu.cn (L.S.); liuleian@163.com (L.L.)

**Keywords:** *Acyrthosiphon pisum*, free amino acids, alfalfa, amino acid transporter, gene functional verification

## Abstract

**Simple Summary:**

Pea aphids (*Acyrthosiphon pisum*) serve as a model insect for ecological research and have recently emerged as a significant pest of alfalfa. Amino acid uptake and balance in insects typically rely on amino acid transporters. In this investigation, RT-qPCR was employed to examine the distinct gene expression patterns of seven amino acid transporters subsequent to pea aphid feeding on resistant and susceptible alfalfa varieties. RNA interference targeting the pea aphid *ACYPI000536* and *ACYPI004320* genes was carried out via a plant-mediated approach, elucidating the preliminary functions of these genes. The study revealed that down-regulating the *ACYPI000536* gene led to increased histidine and lysine levels in pea aphids, subsequently elevating mortality post-feeding on the susceptible alfalfa variety “Lie Renhe”. Conversely, down-regulating the *ACYPI004320* gene resulted in elevated phenylalanine levels in pea aphids, leading to reduced mortality following feeding on the highly resistant alfalfa variety “Gannong 5”.

**Abstract:**

In recent years, pea aphids have become major pests of alfalfa. Our previous study found that “Gannong 5” is a highly aphid-resistant alfalfa variety and that “Lie Renhe” is a susceptible one. The average field susceptibility index of “Gannong 5” was 31.31, and the average field susceptibility index of “Lie Renhe” was 80.34. The uptake and balance of amino acids in insects are usually dependent on amino acid transporters. RT-qPCR was used to detect the relative expression levels of seven amino acid transporter differential genes in the different instar pea aphids fed on resistant and susceptible alfalfa varieties after 24 h, and two key genes were selected. When pea aphids fed on “Gannong 5”, the expression of *ACYPI004320* was significantly higher than that in pea aphids fed on “Lie Renhe”; however, the expression of *ACYPI000536* was significantly lower than that in pea aphids fed on “Lie Renhe”. Afterward, the RNA interference with pea aphid *ACYPI000536* and *ACYPI004320* genes was performed using a plant-mediated method, and gene function was verified via liquid chromatography–mass spectrometry and pea aphid sensitivity to aphid-resistant and susceptible alfalfa varieties. The results showed that the down-regulation of the *ACYPI000536* gene expression led to an increase in the histidine and lysine contents in pea aphids, which, in turn, led to an increase in mortality when pea aphids fed on the susceptible variety “Lie Renhe”. The down-regulation of the *ACYPI004320* gene expression led to an increase in phenylalanine content in pea aphids, which, in turn, led to a decrease in mortality when pea aphids fed on the resistant variety “Gannong 5”.

## 1. Introduction

The pea aphid, scientifically known as *Acyrthosiphon pisum* Harris, is a significant agricultural pest exhibiting color polymorphism, characterized by red and green morphs. Both nymphs and adult aphids congregate on the tender parts of plants, causing substantial damage. Infested plants exhibit symptoms such as leaf curling, bud drop, leaf spotting, and subsequent mold growth triggered by the honeydew excreted by the aphids post-feeding. In the United States, pea aphids cause approximately USD 60 million in annual losses [1] while leading to economic losses ranging from 10% to 30% yearly in alfalfa cultivation regions of Northwest China [2].

The use and cultivation of insect-resistant varieties represent environmentally friendly and sustainable measures for controlling pests in alfalfa [3]. The alfalfa variety “Gannong 5” demonstrates resistance to aphids and thrips, with an average field susceptibility index of 31.31 [4]. Pea aphid mortality was 53.3% after 24 h of feeding on “Gannong 5”, which is 3.63 times higher than the 14.7% mortality at 12 h [5]. Conversely, the “Lie Renhe” alfalfa variety is susceptible to aphids and thrips, exhibiting an average field susceptibility index of 80.34 [4]. The mortality of the pea aphid reached 34.0% after 72 h of feeding on “Lie Renhe” [5].

Insects are unable to independently synthesize certain amino acids and must obtain them from dietary or alternative sources [6,7,8,9,10]. Consequently, plant phloem amino acids are widely considered the primary limiting factors affecting the growth and development of aphids [8]. Amino acids cannot freely traverse cellular membranes; they rely on specific amino acid transporters for translocation. As a result, the acquisition and equilibrium of amino acids in insects are contingent upon the functionality of these transporters. These transporters are not only involved in nutrient absorption but also play crucial roles in various physiological processes, including nervous system regulation and material transport [9].

Known amino acid transporters operate based on electrochemical potential and are typically associated with ion movement. For instance, within the lepidopteran larval midgut, an apical H^+^ V-ATPase and an apical K^+^/2H^+^ antiporter actively facilitate the secretion of K^+^, thereby propelling the transportation of amino acids [9,10,11,12,13]. In other insect species, amino acid transport mechanisms are primarily energized by the sodium motive force [7,14]. The amino acid transport (AAT) families, belonging to the amino acid/polyamine/organocation (APC) superfamily, represent the primary nutritional AATs in eukaryotes and play pivotal nutritional roles in insects [11,12,13,14,15,16,17,18,19,20,21].

Our laboratory previously performed a transcriptome analysis on green pea aphids subsequent to 24 h feeding periods on both resistant and susceptible varieties of alfalfa, resulting in the identification of numerous differentially expressed genes associated with amino acids, notably encompassing seven amino-acid-transporter-related genes. Specifically, *ACYPI004320*, *ACYPI000333*, *ACYPI000536*, *ACYPI000092*, and *ACYPI001684* belong to the amino acid/auxin permease (AAAP) family, while *ACYPI009985* and *ACYPI005156* are affiliated with the amino acid/polyamine/organocation (APC) family. The primary role of the AAAP family involves transporting auxin and one or more amino acids, typically comprising 400–700 amino acid residues in insects [22]. Conversely, amino acid/polyamine/organocation (APC) transporters act as symporters and antiporters, with APC family proteins generally ranging from 350 to 800 amino acid residues in length [22].

Based on the above background and analysis, we propose a hypothesis: After pea aphids feed on the resistant alfalfa variety “Gannong 5”, these pea aphids will not grow and develop normally due to its effect on the amino-acid-transporter-related genes of the pea aphid. The effects of “Gannong 5” on the amino-acid-transporter-related genes of the pea aphid mainly resulted in gene up-regulation and down-regulation. If “Gannong 5” leads to gene down-regulation in the pea aphid, we are curious about the effect of the corresponding gene down-regulation on the pea aphid when it feeds on the susceptible alfalfa variety “Lie Renhe”. Similarly, if “Gannong 5” causes up-regulation of a gene in the pea aphid, we want to investigate the effect on the pea aphid when the corresponding gene is not up-regulated via RNA interference. Therefore, we selected an up-regulated gene with a large fold change and a down-regulated gene with a large fold change in the differential genes by qPCR for gene function verification.

This study utilized RNA interference (RNAi) to target the *ACYPI000536* and *ACYPI004320* genes in pea aphids via a plant-mediated delivery method. Validation of the functionality of *ACYPI000536* and *ACYPI004320* involved assessing amino acid contents using liquid chromatography–mass spectrometry and conducting sensitivity tests on RNAi-treated pea aphids against the susceptible alfalfa variety.

## 2. Materials and Methods

### 2.1. Insect Culture

The pea aphids utilized in this study were sourced from the Insect Ecology Laboratory of the Plant Protection College, Gansu Agricultural University. Single pea aphids were cultured on indoor potted broad bean plants within a controlled light incubator environment (T = 20 ± 1 °C, RH = 70 ± 5%, L:D = 16:8). Following a 24 h reproductive cycle, adult aphids were removed after being placed on clean broad beans. For the experimental procedures, 1st to 2nd instar nymphs, 3rd instar nymphs, 4th instar nymphs, and adult aphids were selectively chosen.

### 2.2. Plant Cultivation

“Gannong 5” and “Lie Renhe” alfalfa varieties were provided by the Insect Ecology Laboratory of College of Plant Protection, Gansu Agricultural University. “Lie Renhe” broad beans are Chinese native varieties, and “Gannong 5” are bred by mass selection [23]. Broad beans were sown in seedling cups (d = 10 cm, h = 14 cm) and raised in a light incubator (T = 25 ± 1 ° C, RH = 70 ± 5 %, L: D = 16L: 8D). Alfalfa (45 days old) and broad beans (7 days old) were used in this study.

### 2.3. Differential Gene Expression Analysis of Amino Acid Transporter in Pea Aphids

Pea aphids of different instars were inoculated on “Gannong 5” and “Lie Renhe” for 24 h, three biological replicates were performed for each treatment, and 30 aphids per replicate were used. After 24 h of treatment, 10 aphids per replicate were sampled and stored in a fridge at −80 °C after liquid nitrogen treatment.

The samples were taken out from the −80 °C fridge with different treatments and placed in the nuclease-free centrifuge tube. Total RNA was extracted using TRIzol reagent (Invitrogen, Carlsbad, CA, USA) following the manufacturer’s protocols. The quality of the RNA samples was measured with 1% agarose gel electrophoresis and then measured using a NanoPhotometer-NP80 spectrophotometer (GE Healthcare, Wiesbaden, Germany). The OD260/280 absorption ratio was between 1.80 and 2.10, which indicated that samples were of acceptable RNA quality and purity for analyses. Then, a total of 1000 ng RNA from each sample was reverse transcribed into cDNA with a Prescripts RT regent Kit and a gDNA Eraser (Takara, Dalian, China). Finally, the cDNAs were stored in a −20 °C refrigerator.

### 2.4. Quantitative Real-Time PCR Analysis of Relative Expression Levels

The relative mRNA expression level was analyzed with RT-qPCR. The primers used are listed in Table 1. The RT-qPCR was carried out using the SYBR Green I chimeric fluorescence method on the fluorescent quantitative PCR instrument (Thermo Fisher Scientific, Singapore). The RT-qPCR reaction was performed with a NoVo Start SYBR qPCR Supermax Plus kit (Oncoprotein Scientific Inc., Shanghai, China), using 0.5 µL cDNA as a template, 5 µL of 2 × NovoStart^®^ SYBR qPCR SuperMix Plus, 0.5 µL of 10 μM Sense-Primer, 0.5 µL of 10 μM AntiSense-Primer, 0.2 µL of ROX Reference DyeII (ROXII), and 3.3 µL of RNase free water. A thermocycler was programmed with the following cycling conditions: (1) 95 °C for 2 min; (2) 40 cycles of 95 °C for 15 s, 60 °C for 30 s and 95 °C for 15 s; and (3) 60 °C for 1 min, 95 °C for 15 s. Ribosomal protein L7 (RPL7) and ribosomal protein S20 (RPS20) were used as the reference genes.

The relative mRNA expression level was defined as the fold change relative to the average expression of *RPL7* and *RPS20*. Relative quantification was calculated using the 2^−ΔΔCt^ method. The algorithm of ΔΔCt was slightly modified as follows: ΔΔCt = ΔCt (Treatment) − ΔCt (Control). The significance was analyzed using the independent sample T test.

### 2.5. Synthetic dsRNA

While screening for gene expression variations resulting from pea aphid feeding on both resistant and susceptible alfalfa varieties, two genes were identified with differing expression levels. Subsequently, the synthesis of double-stranded RNA (dsRNA) involved amplifying two fragment templates, *ACYPI000536* and *ACYPI004320*, via PCR. These fragments were derived from cDNAs previously cloned as templates using primers containing the T7 promoter sequence at their 5′ ends (see Table 2). The dsRNA synthesis was conducted utilizing the Thermo Scientific TranscriptAid T7 High Yield Transcription Kit (Thermo Scientific, Wilmington, MA, USA). Following synthesis, the concentration of the produced dsRNA was measured using a NanoPhotometer-N50 spectrophotometer (GE Healthcare, Wiesbaden, Germany).

### 2.6. RNA Interference

The plant-mediated delivery of dsRNA followed the method outlined by Ye et al. [24]. A 7-day-old broad bean was trimmed 1 cm above the soil, and a truncated incision was made at the rhizome. The bean was then inserted into a 200 μL centrifuge tube with its cap removed. Subsequently, the gap between the bottle mouth and the rhizome was sealed using paraffin film, and dsRNA was injected into the centrifuge tube until full. The dsGFP (Green fluorescent protein) was utilized as the negative control group, while water, instead of RNAi, served as the blank control group. Following this, first instar nymphs were extracted from the broad bean seedlings and transferred to a Petri dish. The mortality count was conducted every 12 h, with surviving aphids at 48 h after dsRNA delivery being collected, treated with liquid nitrogen, and stored at −80 °C. Each treatment involved 60 aphids and three biological replicates. Statistical analysis was carried out using SPSS 23.0 software. One-way ANOVA was employed to assess significance, and multiple comparisons were executed using the S-N-K method.

### 2.7. Determination of the Sensitivity of Pea Aphids to “Gannong 5” and “Lie Renhe” Alfalfa after RNA Interference

After 48 h of RNA interference, the remaining pea aphids were transferred to “Gannong 5” and “Lie Renhe” alfalfa variety for feeding. The mortality count of pea aphids was observed every 12 hours and recorded for 72 hours. For each treatment, 30 aphids and three biological replicates were set up. The data were statistically analyzed using SPSS 23.0 software. One-way analysis of variance significance test was carried out using the S-N-K method for multiple comparisons.

### 2.8. Determination of Free Amino Acid Content of Pea Aphids after RNA Interference

The pea aphids were collected from each RNAi treatment with three biological replicates. From each replicate, five pea aphids were extracted, treated with liquid nitrogen, and then stored in a freezer at −80 °C for the determination of free amino acid content.

#### 2.8.1. Sample Preparation

According to Zhu’s method [25], 5 pea aphids were placed in a 1.5 mL centrifuge tube, weighed, and then agitated by vortexing with 50 μL of a 5% acetic acid solution for 1 minute. Subsequently, they were combined with 950 μL of 5% acetic acid solution. After standing for 30 min, the mixture was centrifuged at 18,000× *g* for 10 min. The supernatant was filtered with 0.22 μm aqueous phase needle filter (Anpel SCAA-102, Shanghai, China).

#### 2.8.2. Determination of Free Amino Acids

According to the Chinese national standard GBT30987-2020 [26], the determination of free amino acid content was carried out using liquid chromatography–mass spectrometry (Agilent 1260 Infinity, Waldbronn, Germany).

##### Flow Phase Configuration

First, 12.612 g of ammonium formate was weighed and dissolved in 1000 mL of water, and the pH value was adjusted to 3.0 with formic acid to prepare a 200 mmol/L ammonium formate solution. Then, 100 mL of the prepared 200 mmol/L ammonium formate solution was weighed, added to 900 mL of water, and fully mixed to make mobile phase A. Next, 100 mL of the 200 mmol/L ammonium formate solution was weight, adding 900 mL acetonitrile, and fully mixed to make mobile phase B.

##### Instrument Conditions

Liquid chromatography conditions are as follows:(1)Chromatographic column: HILIC-Z, 2.7 μm, 2.1 mm × 150 mm, or chromatographic column with the same performance;(2)The specific elution procedure of mobile phase is shown in the Table 3.

(3)Flow rate: 0.5 mL/min;(4)Column temperature: 25 °C; injection volume: 1 μL.

Mass spectrometry conditions are as follows:(1)Scanning mode: positive ion mode;(2)Detection method: multiple reaction monitoring (MRM);(3)Drying gas (N2) temperature: 230 °C;(4)Dry gas flow rate: 11.0 L/min;(5)Sheath gas temperature: 390 °C;(6)Sheath gas flow rate: 12.0 L/min;(7)Atomizing gas (N2) pressure: 0.14 MPa (20 psi);(8)Capillary voltage: 1500 V.

## 3. Results

### 3.1. Effects of Resistant and Susceptible Alfalfa Varieties on Differential Genes of Amino Acid Transporters in Pea Aphids

As shown in Figure 1, when pea aphids fed on “Gannong 5”, the expression of the *ACYPI009985* gene was significantly up-regulated in 1–2 instar nymphs (t_9985_ = 3.534, *p* < 0.05), and the fold change was 1.49. The expression of the *ACYPI004320* gene was significantly up-regulated in the fourth instar nymphs (t_4320_ = 3.202, *p* < 0.05), and the fold change was 1.98.

The expression of *ACYPI000092* and *ACYPI001684* genes was significantly down-regulated in 1–2 instar nymphs (t_0092_ = −3.439, t_1684_ = −3.028, *p* < 0.05), and the fold change was 0.56 and 0.57, respectively. The expression of the *ACYPI000536* gene was significantly down-regulated in the third instar nymphs (t_0536_ = −11.76, *p* < 0.01), showing a fold change of 0.24. In the fourth instar nymphs, both *ACYPI000536* and *ACYPI001684* genes were significantly down-regulated (t_0536_ = −3.074, t_1684_ = −3.55, *p* < 0.05), with fold changes of 0.61 for both.

### 3.2. Effect of RNA Interference on the Survival of Pea Aphids

As shown in Figure 2, the *ACYPI000536* and *ACYPI004320* genes were interfered with using the plant-mediated method, and the interference efficiency of the target genes was detected after 48 h. The results showed that the expression of the *ACYPI000536* gene decreased by 40% after 48 h of interference compared with the dsGFP (*p* < 0.05), indicating that the plant-mediated method could successfully interfere with the *ACYPI000536* gene of the pea aphid. After 48 h of interference, the expression of the *ACYPI004320* gene was decreased by 41% compared with the dsGFP (*p* < 0.05), indicating that the plant-mediated method could successfully interfere with the *ACYPI004320* gene of the pea aphid.

As shown in Table 4, the mortality of pea aphid *ACYPI004320* interfered with using the plant-mediated method for 24 h was significantly different from that of water, the ds*GFP*, and ds*ACYPI000536* (*p* < 0.05). The mortality of pea aphid ds*ACYPI000536* and ds*ACYPI004320* at 36 h was significantly different between water and the ds*GFP* (*p* < 0.05).

The mortality of ds*ACYPI000536* and ds*ACYPI004320* at 12 h and 48 h was not significantly different from that of other treatment groups (F_12h_ = 1.11, F_48h_ = 3.85 *p* > 0.05). The mortality of ds*ACYPI000536* and ds*ACYPI004320* at 48 h was 8.33% and 8.89%, respectively.

### 3.3. Determination of Free Amino Acid Content of Pea Aphids after RNA Interference

The results are shown in Table 5. Interference with the *ACYPI000536* gene of the pea aphid can lead to an increase in the content of asparagine, histidine, and lysine in pea aphids compared with the blank control group (*p* < 0.05), and the difference fold changes are 1.68, 1.70, and 1.30, respectively.

Interfering with the *ACYPI000536* gene of the pea aphid could lead to the decrease in tyrosine, alanine, serine, and total amino acid content in pea aphids (*p* < 0.05), and the difference fold changes were 0.67, 0.76, 0.45, and 0.86, respectively.

After interfering with the *ACYPI000536* gene of the pea aphid, there was no significant difference in the content of other amino acids among water, the ds*GFP*, and ds*ACYPI000536* (*p* > 0.05).

Results in Table 6 show that interfering with the *ACYPI004320* gene of the pea aphid led to an increase in the content of phenylalanine and asparagine in pea aphids compared with the blank control group (*p* < 0.05), and the difference fold changes are 1.18 and 1.37, respectively.

Interfering with the *ACYPI004320* gene of the pea aphid can cause the content of tyrosine, alanine, serine, and total amino acids in pea aphids to decrease compared with the blank control group (*p* < 0.05), and the difference fold changes are 0.74, 0.31, 0.45, and 0.84, respectively. After interfering with the *ACYPI004320* gene of the pea aphid, there was no significant difference in the content of other amino acids among water, the ds*GFP*, and ds*ACYPI004320* (*p* > 0.05).

### 3.4. Determination of Pea Aphid Sensitivity to Susceptible Alfalfa Varieties after RNA Interference

#### 3.4.1. The Sensitivity of Pea Aphids to “Lie Renhe” Alfalfa after Interfering with *ACYPI000536* Was Determined

The results are shown in Figure 3. After interfering with the *ACYPI000536* gene of the pea aphid fed on “Lie Renhe” alfalfa, the mortality increased significantly between 12 and 24 h and slowly after 24 h. The highest mortality was 73.33% at 72 h. The mortality of the pea aphid *ACYPI000536* gene was significantly different from that of CK and the ds*GFP* after 12 h of interference and inoculation to “Lie Renhe” alfalfa (*p* < 0.05).

#### 3.4.2. The Sensitivity of Pea Aphids to “Gannong 5” Alfalfa after Interfering with *ACYPI004320* Was Determined

The results are shown in Figure 4. After interfering with the *ACYPI004320* gene of the pea aphid fed on “Gannong 5” alfalfa, the mortality increased slowly. The highest mortality was 21.67% at 72 h. The mortality of pea aphid *ACYPI004320* was significantly different from that of CK and the ds*GFP* after 12 h of interference with the pea aphid *ACYPI004320* gene and inoculation to the “Gannong 5” alfalfa variety (*p* < 0.05).

## 4. Discussion

There have been numerous prior studies on amino acid transporters. For instance, Fu et al. identified *LdNAT1*, a nutritional amino acid transporter in the NSS family of the potato beetle. In their research, inhibiting *LdNAT1* expression using RNA interference resulted in reduced levels of cysteine, methionine, histidine, phenylalanine, isoleucine, leucine, and serine in the larvae, along with increased concentrations of these amino acids in the larval feces [19]. Our results showed that interference with two AAAP family genes can also lead to changes in some amino acid contents of the pea aphid, which is similar to the results of Fu et al.

The results of this study revealed that interfering with the *ACYPI000536* gene of the pea aphid could result in an increase in asparagine, histidine, and lysine content, while the contents of tyrosine, alanine, serine, and total amino acids decreased. Similarly, interference with the *ACYPI004320* gene in the pea aphid led to an increase in phenylalanine and asparagine content, as well as a decrease in tyrosine, alanine, serine, and total amino acid content. Given that these two genes belong to the AAAP family, and the content of asparagine significantly increased following interference, while the contents of tyrosine, alanine, serine, and total amino acids decreased, it is speculated that silencing the *AAAP* family genes of pea aphids can lead to an increase in asparagine content and a decrease in tyrosine, alanine, serine, and total amino acid content. This phenomenon might arise from a negative feedback regulatory mechanism controlling amino acid metabolism and utilization in insects. Elevated levels of specific amino acids trigger this regulatory mechanism, thereby inhibiting pertinent metabolic pathways or stimulating catabolic processes, resulting in reduced levels of other amino acids.

The interference with genes *ACYPI000536* and *ACYPI004320* resulted in a substantial increase in the content of asparagine in pea aphids and a marked decrease in tyrosine, alanine, serine, and total amino acids. The survival rate of pea aphids significantly increased after being fed on the susceptible variety “Lie Renhe” when interfering with the *ACYPI000536* gene, while it decreased significantly after being fed on the resistant variety “Gannong 5” when interfering with the *ACYPI004320* gene. Therefore, it was concluded that the changes in asparagine, tyrosine, alanine, serine, and total amino acid contents did not affect the survival rates of pea aphids after feeding on resistant and susceptible alfalfa varieties.

Interfering with the *ACYPI004320* gene of the pea aphid can increase the content of phenylalanine in pea aphids and reduce the mortality of the pea aphid after feeding on resistant alfalfa variety “Gannong 5”. This is similar to some early research results showing that increasing the content of nutrients in insect diets is beneficial to insects [27]. This may be due to the specific toxic or regulatory role of phenylalanine in insect physiological metabolism.

Nevertheless, nutrient intake by insects does not consistently yield beneficial outcomes. Kyung-Jin Min’s findings indicate that males of *Culex quinquefasciatus* may suffer significantly adverse consequences due to their association with females after consuming high-nutrient foods in the larval phase [28], suggesting that even if insects can acquire more nutrients, other potential factors may also contribute to increased mortality. Similarly, Paoli’s results demonstrated that a diet mainly consisting of essential amino acids leads to a decrease in the survival rate of worker bees, while a diet solely composed of carbohydrates incurs no cost [29]. Our findings demonstrate that disrupting the *ACYPI000536* gene of pea aphids can increase the content of histidine and lysine in pea aphids, consequently raising pea aphid mortality after feeding on the susceptible variety “Lie Renhe”. This is potentially because excessive histidine and lysine content may interfere with the balance of amino acids in the pea aphid, making it inconsistent with the demand for amino acid ratios at different developmental stages, thereby affecting its normal physiological activities.

## 5. Conclusions

In this study, RT-qPCR was used to detect the relative expression of seven amino-acid-transporter-related differential genes in first to second instar, third instar, and fourth instar pea aphids and adult aphids after feeding on resistant and susceptible alfalfa varieties for 24 h. Two putative key genes were screened out.

RNA interference with *ACYPI000536* and *ACYPI004320* genes of the pea aphid was carried out using the plant-mediated method. It was found that the plant-mediated method could successfully interfere with *ACYPI000536* and *ACYPI004320* genes of pea aphid nymphs. Interfering with the *ACYPI000536* gene of the pea aphid can increase the content of asparagine, histidine, and lysine in pea aphids; however, the contents of tyrosine, alanine, serine, and total amino acids were decreased. Interference with the *ACYPI004320* gene in pea aphids leads to the increase in phenylalanine and asparagine content in pea aphids. It can also lead to the decrease in tyrosine, alanine, serine, and total amino acid content. After interference with the pea aphid *ACYPI000536* gene and infecting the “Lie Renhe” alfalfa variety, the 72 h mortality was 73.33%. After interference with the pea aphid *ACYPI004320* gene and infecting the “Gannong 5” alfalfa variety, the 72 h mortality was 21.67%. The suppression of the *ACYPI000536* gene in the pea aphid increases the levels of histidine and lysine, resulting in higher mortality when the aphids feed on the susceptible alfalfa variety “Lie Renhe”. Conversely, suppressing the *ACYPI004320* gene in the pea aphid increases the phenylalanine levels and decreases aphid mortality following feeding on the resistant alfalfa variety “Gannong 5”.

## Figures and Tables

**Figure 1 insects-15-00020-f001:**
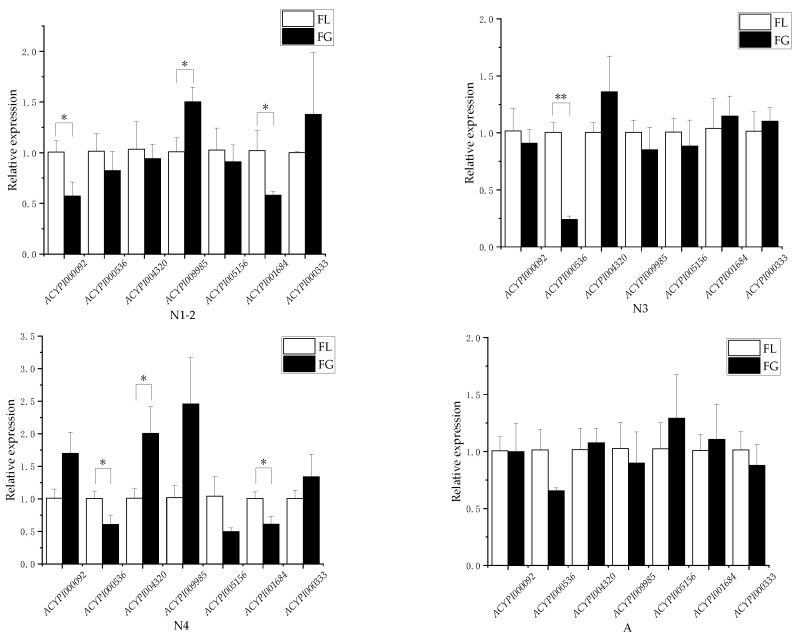
Differential gene expression of amino acid transporters. Note: FG means the pea aphid fed on “Gannon 5”, FL means the pea aphid fed on “Lie Renhe”. * and ** represent significant difference (*p* < 0.05) or extremely significant difference (*p* < 0.01) between FG and FL.

**Figure 2 insects-15-00020-f002:**
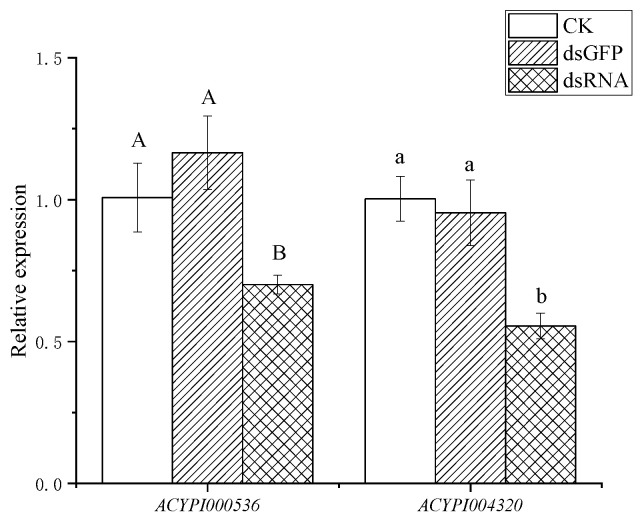
The changes in *ACYPI000536* and *ACYPI004320* gene expression in pea aphids after 48 h of gene interference. Note: CK represents pea aphids mediated by water treatment; dsGFP represents pea aphids that mediate dsGFP treatment; dsRNA represents pea aphids that mediate dsRNA treatment. One-way analysis of variance was used for the significance test. All data are expressed as mean ± standard error. Different capital letters and lowercase letters in the figure indicate that the difference between the target gene and the control after interference was significant (*p* < 0.05).

**Figure 3 insects-15-00020-f003:**
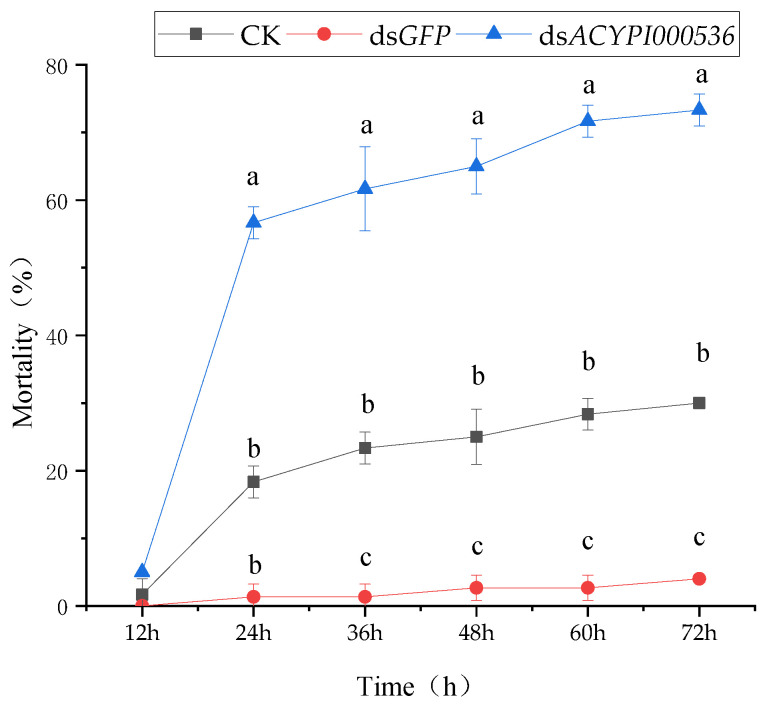
Mortality of pea aphids feeding on “Lie Renhe” alfalfa after interfering with *ACYPI000536.* Note: Different lowercase letters within each time point indicate that there is a significant difference in mortality between different treatments (*p* < 0.05).

**Figure 4 insects-15-00020-f004:**
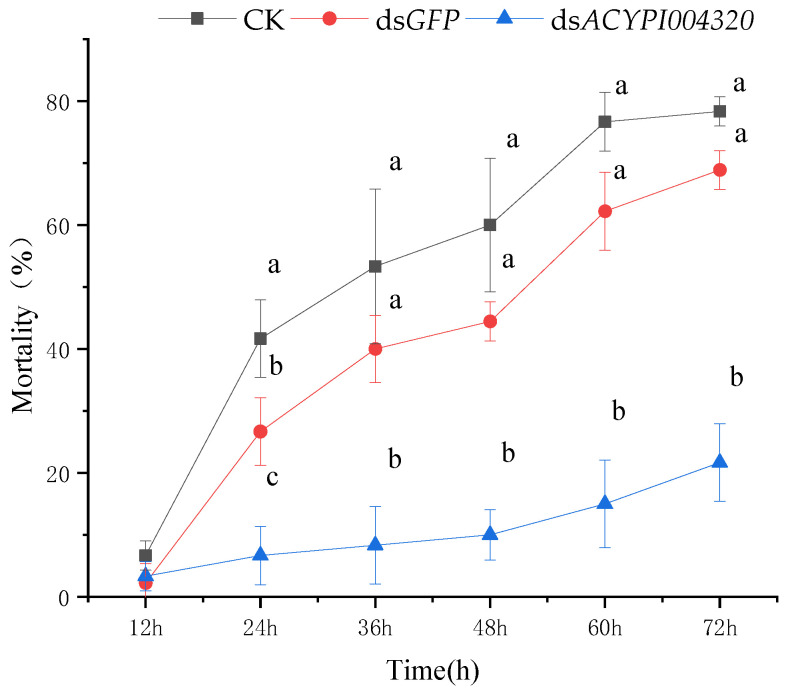
Mortality of pea aphids feeding on “Gannong 5” alfalfa after interfering with *ACYPI004320.* Note: Different lowercase letters within each time point indicate that there is a significant difference in mortality between different treatments (*p* < 0.05).

**Table 1 insects-15-00020-t001:** Real-time fluorescence quantitative primer.

Primer Name	Forward Primer	Reverse Primer
*RPS20*	AAGTGTGTGCTCCGAGATGA	CAGCAATGACACCGGGTTC
*RPL 7*	TTGAAGAGCGTAAGGGAACT	TATTGGTGATTGGAATGCGTTG
*ACYPI004320*	GTTAGTACTAGCCGGTAAC	GTACTATCTCGGTAAGTGC
*ACYPI000333*	CACTCTTGTCCTCCTAAC	GATATCAGAGGGTAGACAG
*ACYPI000536*	GCTTCACTCGCTATATTTGC	TTTATCAACAGGAACGCTGA
*ACYPI000092*	GTCTTACCATTAGAGCAGG	CCGGTAAGAACTATGAGAC
*ACYPI001684*	CAAATGAGTCAGAGTCTCG	GGATGACAAGTCCAAGATG
*ACYPI009985*	CACCACTACTACTACCAC	CAGTCAATCTAGTCTTCCC
*ACYPI005156*	GCTGATCCTCAGGTATAG	GAGAGTCCCAGTTTACTAC

**Table 2 insects-15-00020-t002:** Synthesis of primers for aphid dsRNA.

Primer Name	Forward Primer	Reverse Primer
*dsGFP*	TAATACGACTCACTATAGGG CAGTTCTTGTTGAATTAGATG	TAATACGACTCACTATAGGG GATTTTGGTTTGTCTCCCATG
*dsACYPI000536*	TAATACGACTCACTATAGGG TAGCCTTACCGATTCTTC	TAATACGACTCACTATAGGG GAACCTTCAGTGGAATTG
*dsACYPI04320*	TAATACGACTCACTATAGGG CCGTGTTCTACTTCATTG	TAATACGACTCACTATAGGG ATTATGGAGCCGAGTATC

**Table 3 insects-15-00020-t003:** Specific elution procedure of mobile phase.

Time/min	Mobile Phase	
	A/%	B/%
0	0	100
11.5	30	70
12	0	100
30	0	100

**Table 4 insects-15-00020-t004:** Effect of RNAi on the mortality of pea aphids.

Sample	Mortality/%			
	12 h	24 h	36 h	48 h
CK	1.67% ± 0.0136a	2.78% ± 0.0079b	2.78% ± 0.0079c	6.11% ± 0.0079a
ds*GFP*	2.22% ± 0.0079a	3.33% ± 0.0000b	3.33% ± 0.0000c	5.56% ± 0.0079a
ds*ACYPI000536*	1.11% ± 0.0079a	2.22% ± 0.0079b	5.56% ± 0.0079b	8.33% ± 0.0136a
ds*ACYPI004320*	2.78% ± 0.0079a	5.56% ± 0.0079a	7.22% ± 0.0079a	8.89% ± 0.0157a

Note: CK represents pea aphids treated with water; ds*GFP* represents pea aphids that mediate ds*GFP* treatment; ds*ACYPI000536* represents the pea aphid that mediates ds*ACYPI000536* treatment. ds*ACYPI004320* represents the pea aphid that mediates ds*ACYPI004320* treatment. One-way analysis of variance was used for the significance test. All data are expressed as average mortality ± standard error, and a difference between different lowercase letters indicates significance (*p* < 0.05).

**Table 5 insects-15-00020-t005:** Amino acid content after silencing pea aphid *ACYPI000536* gene.

Amino Acid	CK (μg/mg)	ds*GFP* (μg/mg)	ds*ACYPI000536* (μg/mg)
Thr	0.3707 ± 0.0233 a	0.3860 ± 0.0097 a	0.3614 ± 0.0183 a
Phe	0.3084 ± 0.0121 a	0.2984 ± 0.0231 a	0.3057 ± 0.0085 a
Leu	0.2781 ± 0.0044 ab	0.2986 ± 0.0150 a	0.2526 ± 0.0067 b
Ile	0.4834 ± 0.0218 a	0.4497 ± 0.0416 a	0.4507 ± 0.0286 a
Asn	0.2001 ± 0.0089 b	0.2107 ± 0.0162 b	0.3357 ± 0.0181 a
Trp	0.3655 ± 0.0071 a	0.3550 ± 0.0317 a	0.3510 ± 0.0182 a
Met	0.3902 ± 0.0144 a	0.3570 ± 0.0236 a	0.3605 ± 0.3989 a
Pro	1.2001 ± 0.0144 a	1.1147 ± 0.1288 a	1.2603 ± 0.1244 a
Val	0.6750 ± 0.0155 a	0.6458 ± 0.0307 a	0.6823 ± 0.0487 a
Tyr	5.2146 ± 0.3226 a	4.9761 ± 0.1097 a	3.5015 ± 0.0047 b
Cys	0.3382 ± 0.0273 a	0.3371 ± 0.0392 a	0.2933 ± 0.0162 a
Ala	0.7501 ±0.0109 a	0.7188 ± 0.0217 a	0.5700 ± 0.0783 b
Gly	0.1348 ± 0.0099 a	0.1215 ± 0.0113 a	0.1560 ± 0.0317 a
Ser	0.5416 ± 0.0344 a	0.5403 ± 0.0336 a	0.2419 ± 0.0191 b
Glu	0.1961 ± 0.0058 a	0.1900 ± 0.0215 a	0.1842 ± 0.0180 a
Asp	0.2669 ± 0.0196 a	0.2529 ± 0.0057 a	0.2442 ± 0.0060 a
His	0.3702 ± 0.0090 b	0.3643 ± 0.0184 b	0.6279 ± 0.0328 a
Cys-Cys	0.2714 ± 0.0424 a	0.2716 ± 0.0048 a	0.2827 ± 0.0280 a
Arg	1.0935 ± 0.1458 a	0.9939 ± 0.1436 a	0.9531 ± 0.0722 a
Lys	0.3850 ± 0.0183 b	0.3538 ± 0.0102 b	0.5002 ± 0.0395 a
Total Content	13.8340 ± 0.4266 a	13.1694 ± 0.4213 a	11.9152 ± 0.5598 b

Note: The content of amino acids is expressed as mean ± SD. Different lowercase letters indicate that the content of free amino acids in pea aphids fed on different host plants was significantly different (*p* < 0.05).

**Table 6 insects-15-00020-t006:** Amino acid content after silencing pea aphid *ACYPI004320* gene.

Amino Acid	CK (μg/mg)	ds*GFP* (μg/mg)	ds*ACYPI004320* (μg/mg)
Thr	0.3707 ± 0.0233 a	0.3860 ± 0.0097 a	0.3956 ± 0.0204 a
Phe	0.3084 ± 0.0121 b	0.2984 ± 0.0231 b	0.3643 ± 0.0172 a
Leu	0.2781 ± 0.0044 a	0.2986 ± 0.0150 a	0.2921 ± 0.0091 a
Ile	0.4834 ± 0.0218 a	0.4497 ± 0.0416 a	0.4926 ± 0.0055 a
Asn	0.2001 ± 0.0089 b	0.2107 ± 0.0162 b	0.2744 ± 0.0323 a
Trp	0.3655 ± 0.0071 a	0.3550 ± 0.0317 a	0.3491 ± 0.0129 a
Met	0.3902 ± 0.0144 a	0.3570 ± 0.0236 a	0.3497 ± 0.0269 a
Pro	1.2001 ± 0.0144 a	1.1147 ± 0.1288 a	1.2373 ± 0.0709 a
Val	0.6750 ± 0.0155 a	0.6458 ± 0.0307 a	0.7482 ± 0.0965 a
Tyr	5.2146 ± 0.3226 a	4.9761 ± 0.1097 a	3.8781 ± 0.1699 b
Cys	0.3382 ± 0.0273 a	0.3371 ± 0.0392 a	0.3374 ± 0.0074 a
Ala	0.7501 ±0.0109 a	0.7188 ± 0.0217 a	0.2300 ± 0.0182 b
Gly	0.1348 ± 0.0099 a	0.1215 ± 0.0113 a	0.1339 ± 0.0160 a
Ser	0.5416 ± 0.0344 a	0.5403 ± 0.0336 a	0.2426 ± 0.0152 b
Glu	0.1961 ± 0.0058 a	0.1900 ± 0.0215 a	0.1935 ± 0.0005 a
Asp	0.2669 ± 0.0196 a	0.2529 ± 0.0057 a	0.3112 ± 0.0580 a
His	0.3702 ± 0.0090 a	0.3643 ± 0.0184 a	0.3755 ± 0.0026 a
Cys-Cys	0.2714 ± 0.0424 a	0.2716 ± 0.0048 a	0.3507 ± 0.0533 a
Arg	1.0935 ± 0.1458 a	0.9939 ± 0.1436 a	0.7504 ± 0.0133 a
Lys	0.3850 ± 0.0183 a	0.3538 ± 0.0102 a	0.3737 ± 0.0238 a
Total Content	13.8340 ± 0.4266 a	13.1694 ± 0.4213 a	11.6803 ± 0.2118 b

Note: The content of amino acids is expressed as mean ± SD. Different lowercase letters indicate that the content of free amino acids in pea aphids fed on different host plants was significantly different (*p* < 0.05).

## Data Availability

The transcriptomic data presented in this study are openly available in NCBI SRA database (https://www.ncbi.nlm.nih.gov/sra/SRX18238383[accn], accessed on 11 November 2022) at project number PRJNA892899.

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
