# Peer review of "Functional Analysis of Amino Acid Transporter Genes ACYPI000536 and ACYPI004320 in Acyrthosiphon pisum"

_insects, 2023, doi:10.3390/insects15010020_

Round 1
Reviewer 1 Report
Comments and Suggestions for Authors
This manuscript conducted functional analysis of two amino acid transporter genes ACYPI000536 and ACYPI004320 in the pea aphid Acyrthosiphon pisum by RNAi. The authors made a main conclusion that the downregulation of ACYPI000536 gene could increase the content of histidine and lysine in pea aphid, which led to the increase of mortality on the susceptible alfalfa variety 17 “Lie Renhe” and the downregulation of ACYPI004320 gene can increase the content of phenylalanine in pea aphid and reduce the mortality of pea aphid on high resistance alfalfa variety “Gannong 5”. The quality of the article is not too bad.
I have a major concern, whether the conclusions answer the question about the relationship between Amino acid Transporter Genes and the resistance of alfalfa variety?
Minor concern or suggestions:
(1) redundant expression
It is better to delete the first sentence in the abstract.
line 24 to 26: express like that --Our previous study found that “Gannong 5” is a high aphid-resistant alfalfa variety and “Lie Renhe” a susceptible one.--
(2) Plant mediated RNAi refers to expression of dsRNA by plant or uptake exogenous ds RNA by plant. If it is the latter, how to prove that dsRNA has been ingested by aphids.
(3) Why did only those few amino acids change, rather than the other amino acids?
Comments on the Quality of English LanguageIt's best to find a native speaker to modify the English expressions.
Reviewer 2 Report
Comments and Suggestions for Authors
ABSTRACTS
-Line 10: Do not start the sentence with “And”. Check this in other parts of the text.
-Lines 15-19: “Pea aphid” is repeated too much.
-Line 29: „whole instars“ - Do you mean „all instars“? According to line 97 you also worked with adults.
-Line 30: “screened” – I suggest “selected”.
-Lines 36-37: Asparagine is also increased.
-Line 39: Asparagine is also increased.
INTRODUCTION
-Line 45: Add “Harris” after “pisum”.
-Line 47: “affected” – I suggest “infested” or “attacked”.
-Lines 71-74: This sentence is confused. Check it for grammar.
-Lines 75-75: Does this refer to results presented in the manuscript? If the answer is yes than “previously” should be replaced with “first”.
-Lines 86-90: Explain here why you selected these two genes and what are the aims and hypotheses of your experiment.
MATERIAL AND METHODS
-Lines 103-104: Space is missing after “Selection”. Later I noticed that many spaces are missing in the text (especially in References). Please, check this in the whole text. In line 104 I noticed comma after full stop. Check the whole text for such mistakes.
-Line 105: Do you mean “broad been seedlings”?
-Lines 181-183: The first sentence is not complete. Merge it with the second sentence.
-Parts 2.9.2.1 and 2.9.2.2 are written as laboratory protocols. I think that you should use sentences to explain the method.
RESULTS
-Line 213: Usually df is presented in subscript of F value. F value of 0.36 cannot be significant. I see in figure that difference is large. So, the 0.36 might not be correct value.
-Expression of two selected genes did not differ between broad bean varietes in 1st-2nd instars when RNA interference was performed. Please, explain why you chose early instars instead of instars when expression of these genes significantly differed.
-Table 3: Put mortality instead of survival in caption to Table 3. You presented 36h before 48h. Although some differences are statistically significant the values of mortalities are quite low. So, 40% decrease in gene expression did not provoke large effects on aphids during first 36h. Later, when they were transferred on broad bean (after 48h of interference) effects are large. This should be discussed.
-Table 4: Can you provide any suggestion in Discussion why some (specific) amino acids are increased and other decreased as a consequence of RNAi.
-Line 273: Start the sentence with “Results in Table 5 show that interfering with the ACYPI004320 gene…”
-Explain lowercase letters in Figures 3 and 4. It is strange that mortality after 72h in control group in Fig. 3 is about 30% and in Fig. 4. It is about 80%. Please, explain. Why you transferred aphids of RNAi 4320 only on “Gannong 5” and RNAi536 only on “Le Renhe”? What was your research question?
DISCUSSION
-The discussion is mostly repeating the results and only 4 references are cited. Role of transporters in aphid survival on different varietes should be explained and some mechanisms should be suggested. How changes in amino acid ratios affect mortality? Why increased histidine and lysine increase mortality on Le Rehne and why increased phenylalanine decrease mortality on Gannong 5?
-Lines 319-321, 324-326: The sentences are incomplete.
-Line 321: It is not speculated. Your results show that it was the consequence of silencing.
-Lines 330-332: However, in Abstract you said that changes in amino acids led to changes in mortality (lines 35-40).
-Lines 338-339: Use italic for D. melanogaster.
-Please, explain better the connection of your results with references 32 and 33.
REFERENCES
-Many spaces are missing.
-Some authors are with full first names.
-Some last names of authors are with all capital letters
-What heppened with author names in references 14, 15,…?
-Volume and/or page numbers are missing in some references.
-Some article titles are with each word capitalized.
-Some species names are with all lowercase letters and/or are not in italic.
-Some journal names are not in italic.
-Some words in article title are splitted without hyphenation.
-Put K+/H+ in reference 13.
-Bmc should be BMC in references 20, 25
-“Et” in reference 24 should be with lowercase letters.
-Check if reference 29 is in accordance with propositions of INSECTS.
Comments on the Quality of English Language
I provided several suggestions in comments to authors. Some sentences are not complete or are confused. Some sentences start with "And".
Round 2
Reviewer 2 Report
Comments and Suggestions for Authors
The manuscript is mostly revised according to my comments and suggestions and I am sattisfied with the answers. The Discussion is improved. I have only few minor comments. However, please, pay attention to my remark regarding the previous comment 16 (see below).
Comment 12: There are still missing spaces (e.g., after reference numbers; line 287). Please, check again.
New lines 115-116: Start sentence with “Alfalfa (45 days old) and broad beans (7 days old) were used…”. The sentence should not start with the number.
New lines 202-203: Change “ml” to “mL”.
Comment 16: It is not possible that such low F value is with PË‚0.05. Check again and see some tables with F values (e.g., https://users.sussex.ac.uk/~grahamh/RM1web/F-ratio%20table%202005.pdf)
Comment 18: Yes, I ment on your results presented in Figs 3 and 4.
Comment 21: In lines 309 and 320 replace “in the same column” with “within each time point”. Column is more appropriate for tables.
REFERENCES:
-Words are splitted between two rows without hyphenation (references 1,4,6,9 etc.).
-Authors are with first names (references 1,2,17,18,24,25).
-Article title is with each word capitalized (references 5,25).
-Page (or article) numbers or volumes are missing. In reference 12 put 2365-2375. In reference 16 put article number 253. In reference 18 put 320-325. In reference 20 put 1848(10), 2085-2091. Are there page numbers for reference 23? In reference 24 article number is 3694. In reference 27 put page numbers 643-646. In reference 28 put 1659-1664.
-You put commas between initials in references 12,13,
-In reference 16 you put sign “&”. Put “Slimfast” with lowercase letters and italic.
-In reference 20 there are 3 coauthors. “Daniel” is not last name, but the first name of Price. So authors should be “Price D. R. G., Wilson A. C. C., Luetje C. W.”.
-In reference 28 put Culex quinquefasiatus in italic.
